https://doi.org/10.1038/s41467-019-08616-0　　**OPEN**

# On the predictability of infectious disease outbreaks

Samuel V. Scarpino[1,2,3,4,5,6] & Giovanni Petri [6,7]

Infectious disease outbreaks recapitulate biology: they emerge from the multi-level interaction of hosts, pathogens, and environment. Therefore, outbreak forecasting requires an integrative approach to modeling. While specific components of outbreaks are predictable, it remains unclear whether fundamental limits to outbreak prediction exist. Here, adopting permutation entropy as a model independent measure of predictability, we study the predictability of a diverse collection of outbreaks and identify a fundamental entropy barrier for disease time series forecasting. However, this barrier is often beyond the time scale of single outbreaks, implying prediction is likely to succeed. We show that forecast horizons vary by disease and that both shifting model structures and social network heterogeneity are likely mechanisms for differences in predictability. Our results highlight the importance of embracing dynamic modeling approaches, suggest challenges for performing model selection across long time series, and may relate more broadly to the predictability of complex adaptive systems.

[1] Network Science Institute, Northeastern University, Boston, MA 02115, USA. [2] Marine & Environmental Sciences, Northeastern University, Boston, MA 02115, USA. [3] Physics, Northeastern University, Boston, MA 02115, USA. [4] Health Sciences, Northeastern University, Boston, MA 02115, USA. [5] Dharma Platform, Washington, DC 20005, USA. [6] ISI Foundation, 10126 Turin, Italy. [7] ISI Global Science Foundation, New York, NY 10018, USA. These authors contributed equally: Samuel V. Scarpino, Giovanni Petri. Correspondence and requests for materials should be addressed to S.V.S. (email: s.scarpino@northeastern.edu) or to G.P. (email: giovanni.petri@isi.it)

" I f we don't have a vaccine–yes, we are all going to get it[1]." This dire assessment by a Canadian nurse in 2003 reflected the global public health community's worst fears about the ongoing severe acute respiratory syndrome (SARS) outbreak[2,3]. These fears—for perhaps the first time in history—were partially derived from mathematical and computational models, which were developed in near real-time during the outbreak to forecast transmission risk[3,4]. However, the predictions for SARS failed to match the data[3,5]. Over the subsequent 15 years, the scientific community developed a rich understanding for how social contact networks, variation in health-care infrastructure, the spatial distribution of prior immunity, etc., drive complex patterns of disease transmission[6–11], and demonstrated that data-driven, dynamic, and or agent-based models can produce actionable forecasts[12–17]. Additionally, studies have demonstrated that predicting different components of outbreaks—e.g., the expected number of cases, pace, and tempo of cases needing treatment, demand for prophylactic equipment, importation probability, etc.—is feasible[3,13,18–24]. Despite these advances, an ongoing debate continues in the scientific community about both the need and our capacity to forecast outbreaks[25,26]. What remains an open question is whether the existing barriers to forecasting stem from gaps in our mechanistic understanding of disease transmission and low-quality data or from fundamental limits to the predictability of complex, sociobiological systems, i.e. outbreaks[4,6,7,27–30].

In order to study the predictability of diseases in a comparative framework, which also permits stochasticity and model non-stationarity, we employ permutation entropy as a model-free measure of time-series predictability[31–33]. This measure, i.e permutation entropy, is ideal because—in addition to being a model independent metric of predictability—recent work has demonstrated that it correlates strongly with known limits to forecasting in dynamical systems, e.g., models where we can measure Lyapunov stability[31–33] and can be transformed into an estimate of Kolmogorov-Sinai entropy[34]. Additionally, recent studies by Pennekamp et al.[33] and Garland et al.[35] demonstrated that permutation entropy correlated strongly with forecast accuracy for ecological models and with anomalies in climatological data.

Studying the predictability of a diverse collection of historical outbreaks—including, chlamydia, dengue, gonorrhea, hepatitis A, influenza, measles, mumps, polio, and whooping cough—we identify a fundamental entropy barrier for infectious disease time-series forecasting. However, we find that for most diseases this barrier to prediction is often well beyond the timescale of single outbreaks, implying prediction is likely to succeed. We also find that the forecast horizon varies by disease and demonstrate that both shifting model structures and social network heterogeneity are the most likely mechanisms for the observed differences in predictability across contagions. Our results highlight the importance of moving beyond time-series forecasting, by embracing dynamic modeling approaches to prediction[36], and suggest challenges for performing model selection across long disease time series. We further anticipate that our findings will contribute to the rapidly growing field of epidemiological forecasting and may relate more broadly to the predictability of complex adaptive systems.

## Results

**Permutation entropy as the predictability of disease time series.** Permutation entropy is conceptually similar to the well-known Shannon entropy[31]. However, instead of being based on the probability of observing a system in a particular state, it utilizes the frequency of discrete motifs, i.e symbols, associated with the growth, decay, and stasis of a time series. For example, in a binary time series the permutation entropy in two dimensions would count the frequency of the set of possible ordered pairs, {[01], [10]}, and the Shannon entropy, or uniformity, of this distribution is the permutation entropy. In higher dimensions, one can define an alphabet of symbols over all factorial combinations of orderings in a given dimension, e.g., {[0, 1, 2],[2, 1, 0],[1, 0, 2], etc.}, over which the permutation entropy will be defined. A time series that visits all the possible symbols with equal frequency will have maximal entropy and minimal predictability, and a time series that only samples a few of the possible symbols will instead have lower entropy and hence be more predictable.

More formally, for a given time series $\{x_t\}_{t=1,...,N}$ indexed by positive integers, an embedding dimension $d$ and a temporal delay $\tau$, we consider the set of all sequences of value $s$ of the type $s = \{x_t, x_{t+\tau}..., x_{t+(d-1)\tau}\}$. To each $s$, we then associate the permutation $\pi$ of order $d$ that makes $s$ totally ordered, that is $\tilde{s} = \pi(s) = [x_{t_i}, ... , x_{t_N}]$ such that $x_{t_i} < x_{t_j} \forall t_i < t_j$, hence generating the symbolic alphabet. Ties in neighboring values, i.e. $x_{t_i} = x_{t_j}$, were broken both by keeping them in their original order in the time series and/or by adding a small amount of noise, the method of tie-breaking did not affect the results, see ref. [37] for more details on tie-breaking and permutation entropy. The permutation entropy of time-series $\{x_t\}$ is then given by the Shannon entropy on the permutation orders, that is $H^p_{d,\tau}(\{x_t\}) = -\sum_\pi p_\pi \log p_\pi$, where $p_\pi$ is the probability of encountering the pattern associated with permutation $\pi$ (see Supplementary Figure 1).

As described above, calculating the permutation entropy of a time series requires selecting values for the embedding dimension $d$, the time delay $\tau$, and the window length $N$ *over which permutation entropy is calculated*. In this study, our goal was to find conservative values of $H^p$ by searching over a wide range of possible $(d, \tau)$ pairs and setting $H^p(\{x_t\}) = \min_{d,\tau} H^p_{d,\tau}(\{x_t\})$. However, the value of $H^p$ should always decline as the embedding dimension $d$ grows, i.e. no minimum value of $H^p$ will exist for finite windows sizes $N$. To address this issue, we follow Brandmaier[38] and exclude all unobserved symbols when calculating $H^p$, which acts as a penalty against higher dimensions and results in a minimum value of $H^p$ for finite length time series. To control for differences in dimension and for the effect of time-series length on the entropy estimation, we normalize the entropy by $\log(d!)$, ensure that each window is greater in length than $d!$, and confirm that the estimate of $H^p$ has stabilized (specifically that the marginal change in $H^p$ as data are added is <1%). To facilitate interpretation, we present results from continuous intervals by fixing $\tau = 1$. However, our results generalize to the case where we fix both $d$ and $\tau$ across all diseases and where we minimize over a range of $(d, \tau)$ pairs (see Supplementary Figure 4).

Permutation entropy does not require the a priori specification of a mechanistic nor generating model, which allows us to study the predictability of—potentially very different—systems within a unified framework. What is not explicit in the above formulation is that the permutation entropy can be accurately measured with far shorter time series than Lyapunov exponents and that it is robust to both stochasticity and monotonous transformations of the data, i.e. it is equivalent for time series with different magnitudes[31,39]. Consider—for example—two opposite cases with respect to their known predictability, pure white noise, and a perfectly periodic signal. We expect the former, being essentially random, to display a very high entropy as compared with the latter, which instead we expect to show a rather low entropy in consideration of its simple periodic structure. In Fig. 1, we demonstrate that this is indeed the case, even when we allow the periodic signal to be corrupted by a small

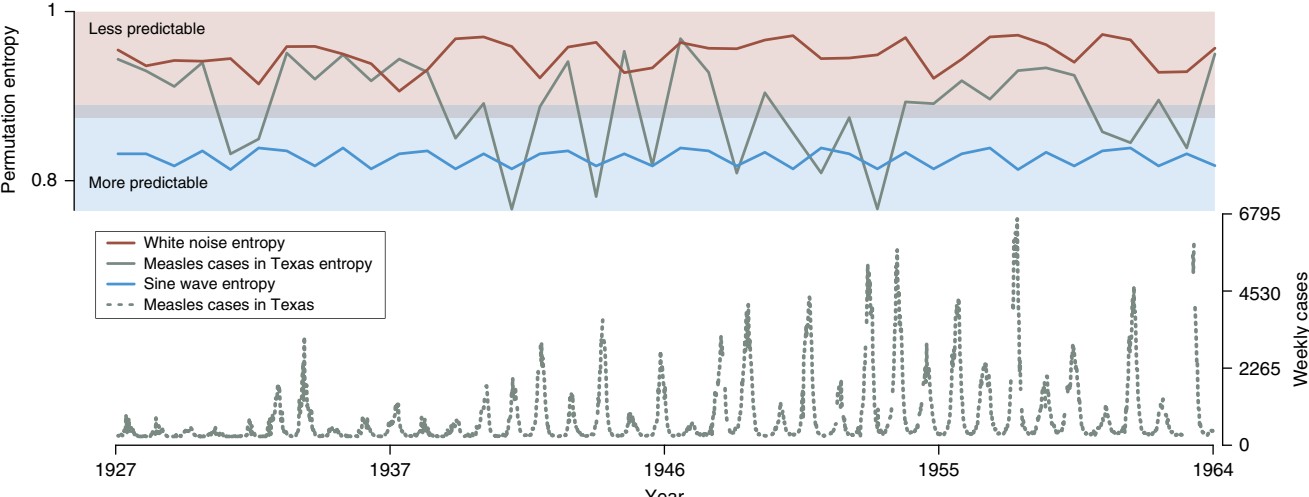

**Fig. 1** Permutation entropy varies through time for real-world disease time series. The permutation entropy in windows of size 52 weeks across three time series of equal length: (1) dark red: Gaussian–white–noise ($\mu = 0$, $\sigma = 1$); (2) gray: measles cases reported to the United States Centers for Disease Control and Prevention from the state of Texas between 1927 and 1965 (gray dashed line, lower panel) as digitized from MMWR reports by Project Tycho[60]; and (3) blue: a sine wave with Gaussian noise ($\mu = 0$, $\sigma = 0.01$). The fluctuations in the permutation entropy calculated from the measles time series (gray dashed line) are larger than would be predicted by chance and result in periods of time where model-based forecasts should be accurate (in the range of a noisy sine wave, blue shaded region) and periods of time (in the range of the white noise, red shaded region) when statistical forecasts based solely on the time-series data should outperform model-based forecasts

amount of noise. We track the short-scale predictability of the time series by calculating the permutation entropy in moving windows (with width = 1 year, although the results are robust to variation in window size). For comparison, we calculate the same moving-window estimate of the permutation entropy for the time series of measles cases in Texas prior to the introduction of the first vaccine. The critical observation is that the moving-window entropy for the measles time series fluctuates between values comparable with that of pure random noise and, at times, values closer to the more predictable periodic signal, which suggests alternating intervals with different dynamical regimes and, thus, predictability. The magnitude of the entropy fluctuations for measles in Texas is statistically significant by permutation test, $p < 0.001$, as compared with simulated fluctuations obtained by building an estimated multinomial distribution over the symbols and repeatedly calculating the expected Jensen–Shannon (JS) divergence from simulations.

**Pathogen-dependent entropy horizons**. We now turn our attention to a broader set of diseases and ask how the predictability, defined as $\chi = 1 - H^p$ (where $H^p$ is the permutation entropy), scales with the amount of available data (i.e. the time-series length). Specifically, we compute the permutation entropy across more than 25 years of weekly data at the US state-level for chlamydia, dengue, gonorrhea, hepatitis A, influenza, measles, mumps, polio, and whooping cough and plot the predictability ($\chi = 1 - H^p$) as a function of the length of each time series. Focusing first on the predictability over short timescales (Fig. 2), for each time series we average $H^p$ over temporal windows of width up to 100 weeks by selecting 1000 random starting points from each state-level time series for disease and calculating $H^p$ for windows of length 10, 12, ..., 100.

We find that all diseases show a clear decrease in predictability with increasing time series length , which implies that accumulating longer stretches of time-series data for a given disease does not translate into improved predictability. However, we also find strong evidence that the majority of single outbreaks—i.e. temporal horizons characteristic for each disease—are predictable. The

confidence intervals in Fig. 2 show that there can be large variation in predictability across outbreaks of the same disease, providing a first indication of the presence of a changing underlying model structures and or dynamics on the scale of months. We obtained similar results, e.g., decreasing predictability with time-series length, clustering of diseases, and the emergence of barriers to forecasting, using a weighted version of the permutation entropy, which reduces the dependence of the standard unweighted permutation entropy on rare, large fluctuations and by considering estimates of the permutation entropy where the time delay, $\tau$, is allowed to vary[32,40] (see Supplementary Figure 4). By comparison, across all models with fixed structures studied to date, e.g., white noise, sine waves, and even chaotic systems, the predictability is constant in time or is expected to improve with increasing amounts of time-series data[41].

Zooming out, what is also conspicuous about the relationship between time-series length and predictability is that diseases cluster together and show disease-specific slopes, i.e. predictability vs. time-series length, which suggests that permutation entropy is indeed detecting temporal features specific to each disease (Fig. 3a). After re-normalizing time for each disease by its corresponding $R_0$ (the average number of secondary infections a pathogen will generate during an outbreak epidemic when the entire population is susceptible, very large, and is seeded with a single infectious individual)—we used the mean of all reported values found in a literature review (see Supplementary Table 1)—we find that the best-fit mixed-effect slope on a log scale is 1 and that the residual effect is well predicted by the times series' embedding dimension $d$ (see Supplementary Figures 2 and 3). Moreover, because the embedding dimension $d$ of a time series is the length of the basic blocks used in the calculation of the permutation entropy, it encodes the fundamental temporal unit of predictability in the form of an entropy production rate, thus implying that predictability decreases with time-series data at a disease-specific rate determined to first order by $R_0$, which is further modulated by $d$. The result that predictability depends on temporal scale also suggests that the permutation entropy could be an approach for justifying the utility of different data sets, i.e. one could determine the optimal granularity of data by selecting the dimension that maximized predictability.

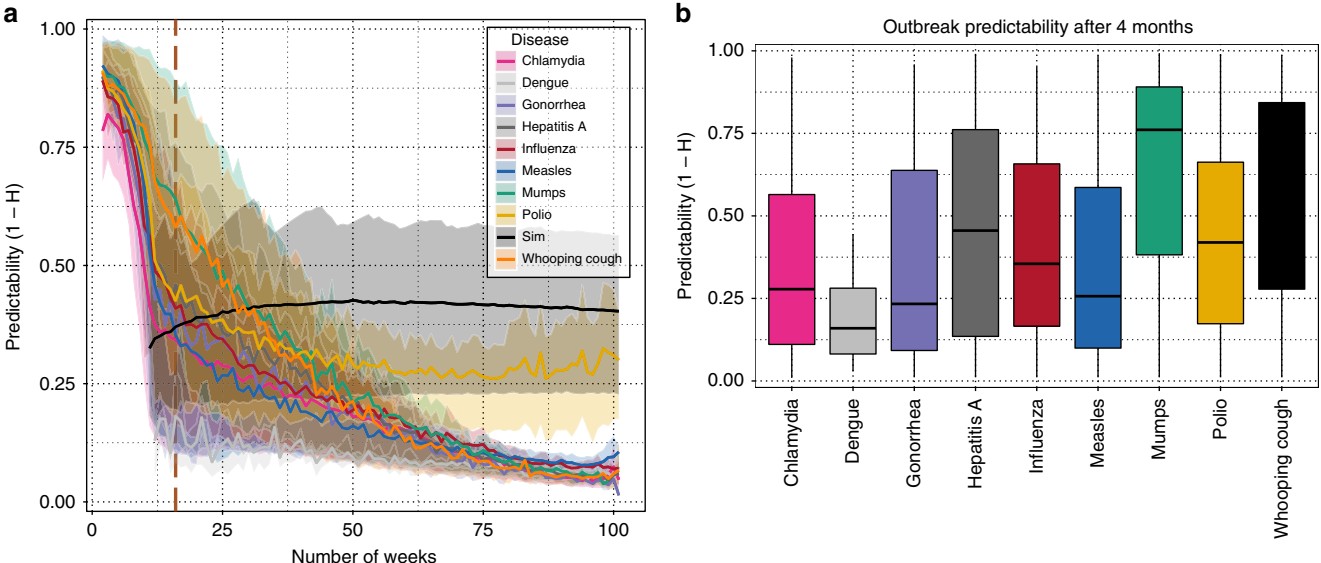

**Fig. 2** Single outbreaks are often predictable. **a** The average predictability $(1 − H^p)$ for weekly, state-level data from nine diseases is plotted as a function of time-series length in weeks. For each disease, we selected 1000 random starting locations in each time series and calculated the permutation entropy in rolling windows in lengths ranging from 2 to 104 weeks. The solid lines indicate the mean value and the shaded region marks the interquartile range across all states and starting locations in the time series. Although the slopes are different for each disease, in all cases, longer time series result in lower predictability. However, most diseases are predictable across single outbreaks and disease time series cluster together, i.e. there are disease-specific slopes on the relationship between predictability and time-series length. To aid in interpretation, the black dashed line plots the median permutation entropy across 20,000 stochastic simulations of a Susceptible Infectious Recovered (SIR) model, as described in the Supplement. This SIR model would be considered predictable, thus values above the black line might be thought of as in-the-range where model-based forecasts are expected to outperform forecasts based solely on statistical properties of the time-series data. The dark brown, dashed vertical line indicates the time period selected for **b**. In **b**, the predictability is shown after 4 months, i.e. 16 weeks, of data for each pathogen. The same procedure was used to generate the permutation entropy as in **a**. The mean predictability differed both by disease and by geographic location, i.e state (analysis of variance with post hoc Tukey honest significant differences test and correction for multiple comparison, sum of squares (SS) disease = 98.22, degrees of freedom (DF) disease = 8, *p*-value disease < 0.001; SS location = 94.7, DF location = 53, *p*-value location < 0.001). The solid line represents the median, boxes enclose the 25th to 75th percentiles of the distributions, and whiskers cover the entire distribution

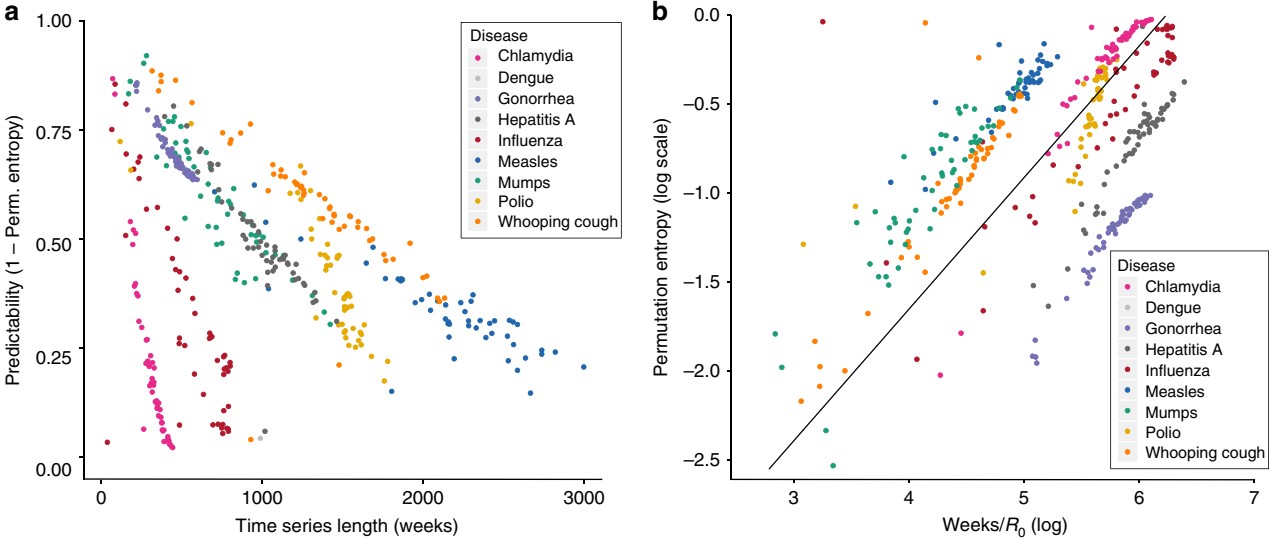

**Fig. 3** Permutation entropy and time-series length show regularity by disease. **a** The predictability $(1 − H^p)$ for chlamydia, dengue, gonorrhea, hepatitis A, influenza, measles, mumps, polio, and whooping cough is plotted as a function of time-series length in weeks. Although the slopes are different for each disease, in all cases, longer time series, i.e. more data, result in lower predictability. However, we again find that single outbreaks should be predictable and that diseases show a remarkable degree of clustering based on the slope of entropy gain. In this analysis, each dot represents the predictability for an entire state-level time series for a disease, i.e. the window size is the entire time series. **b** Rescaled time-series length based on the mean published basic reproductive number. Here we plot the log of time normalized by the basic reproductive number, i.e. $R_O$, from the literature (see Supplementary Table 1) against the log of the permutation entropy

**Drivers of disease time-series predictability**. One might assume that this phenomenon, i.e. decreasing predictability with increasing time-series length, could be driven purely by random walks on the symbolic alphabet used in the permutation entropy estimation. However, $n$-dimensional Markov chain models built from the time-series embeddings ($n = d$ the time-series embedding dimension) consistently produced stable and smaller predictability values in comparison with those obtained from data, corroborating that the predictability behavior we observe does not stem from random fluctuations but is an actual fundamental feature of spreading processes (see Supplementary Figure 6 and Methods for details on the Markov chain simulations). This observation that Markov chain models of the same embedding order do not reproduce the observed predictability indicates that either the model structure is changing in time and/or the system has a very long memory, which is consistent with our current understanding of the entanglement between mobility and disease[3,42]. That the best-fit $n$-dimensional Markov chain models over-predict the amount of entropy in real systems, also supports our earlier results that predictable structure does exist across most long outbreak time series.

To gain insight into what mechanisms might be driving changes in the predictability, we take advantage of the repeated, natural experiment of vaccine introduction. For diseases, such as measles, where we have data from both the pre- and post-vaccine era, we ask whether the permutation entropy changes after the start of widespread vaccination. We consistently observe that predictability decreases after vaccination, again with significance determined by permutation test (Fig. 4a). We also find that the symbol frequency distribution changes significantly after vaccination, as measured by the Jensen–Shannon divergence, across all states in the United States (Fig. 4b). Critically, because—as stated earlier—permutation entropy is not affected by changes in magnitude, the difference in entropy cannot simply be accounted for by a reduction in cases. Instead, it means that the temporal pattern of cases changes after vaccination. This leads us to the hypothesis that the distribution of secondary infections, its first moment or $R_0$ and its higher moments, drives predictable changes in the permutation entropy, a phenomenon originally discovered in synthetic directed networks by Meyers et al.[43].

To further evaluate the hypothesis that heterogeneity in the number of secondary infections produces predictable changes in permutation entropy, we simulate an SIR model with probabilistic restart at end of each outbreak (details in the Supplement) on two classes of temporal networks constructed from the Simplicial Activity Driven (SAD) model[43], a modified activity driven (AD) model in which an activated node contacts $s$ other nodes and induces new links between the contacted nodes (see Methods). In this model we can control the epidemic threshold and the number of secondary contacts by changing the activity and the number of contacted nodes per activation. We simulated two scenarios, one in which the number of contacted nodes per activation is fixed (regular SAD) and one in which we allow fluctuations in contact number (irregular SAD), which generates fluctuations in the number of secondary infections. For both models, we investigated the predictability from below to above the epidemic or critical transmissibility threshold (set to 1 here). From the resulting epidemic curves, we calculated the permutation entropy. Figure 5a shows that we find the same pattern of decreasing predictability observed in real data with longer time series. Figure 5b shows the predictability obtained for the two scenarios below and above the transition: we see that that the strongest difference is present below the transition, where the lack of peculiar structure (the regular contact pattern) induces lower predictabilities than for heterogeneous contact distributions. Above the transitions, we find a reduced effect of the difference in contact structure.

## Discussion

From these results, we can draw three conclusions. First, differences in the average reproductive number, coupled with heterogeneity in the number of secondary infections, can drive differences in predictability across diseases and outbreaks, which is related to results on predicting disease arrival time on networks[44] and to recurrent epidemics in hierarchical meta populations[45]. Second, the permutation entropy could provide a model-free approach for detecting epidemics, which is similar to a recent model-based approach based on bifurcation delays[46–48]. Finally, as outbreaks grow and transition to large-scale epidemics, they should become more predictable, which—as seen in Figs. 1 and 3—appears to be true for real-world diseases as well and agrees with earlier results on how permutation entropy relates to the predictability of nonlinear systems[32].

Our finding that horizons exist for infectious disease forecast accuracy and that aggregating over multiple outbreaks can actually decrease predictability is supported by five additional lines of evidence. First, Hufnagel et al., using data on the 2004 SARS outbreak and airline travel networks, demonstrated that heterogeneity in connectivity can improve predictability[27]. Second, de Cellés et al. noted a sharp horizon in forecast accuracy for whooping cough outbreaks in Massachusetts, USA[49]. Third, Coletti et al. demonstrated that seasonal outbreaks of influenza in France often have unique spatiotemporal patterns, some of which cannot be explained by viral strain, climate, nor commuting patterns[50]. Fourth, Artois et al. found that while it was possible to predict the presence human A(H7N9) cases in China, they were unable to derive accurate forecasts for the temporal dynamics of human case counts[51]. Finally, using state-level data from Mexico on measles, mumps, rubella, varicella, scarlet fever, and pertussis, Mahmud et al. showed evidence that while short-term forecasts were often highly accurate, long-term forecast quality quickly degraded[52].

Research in dynamical systems over the the past 30 years has demonstrated that prediction error increases with increasing forecast length[41]. However, across that same body work, researchers typically find that predictions improve when they are trained on longer time series, even for chaotic systems[41]. Indeed, even for permutation entropy, an active area of research is how spurious aspects of time series can lead to spuriously increasing predictability with increasing time-series lengths[37]. Our data-driven results suggest that for infectious diseases the opposite is true, more time-series data might often lead to lower predictability. Then, by integrating our biological understanding of each pathogen and simulated outbreaks, we found that changing dynamics, e.g., empirical changes in vaccination coverage and simulated shifts in the number of secondary infections as a disease moves through a heterogeneous social network, can cause the prediction error to increase with increasing data, which is related to earlier findings on the role of airline travel networks and disease forecasting[36]. What this implies is that different models generate data at different time points and suggests that the optimal coarse-graining of complex systems might change with scale and or time[53]. The potential for scale-dependent models of infectious disease transmission is supported by a recent analysis of US city-level data on influenza outbreaks that found consistent, mechanistic differences in outbreak dynamics based on city size[54].

The global community of scientists, public health officials, and medical professionals studying infectious diseases has placed a high value on predicting when and where outbreaks will occur, along with how severe they will be[3,27,55,56]. Our results demonstrate that outbreaks should be predictable. However, as outbreaks spread—and spatiotemporally separated waves become entangled with the substrate, human mobility, behavioral changes, pathogen evolution, etc.—the system is driven through a space of diverse model structures, driving down predictability despite increasing time-series lengths. Taken together, our results

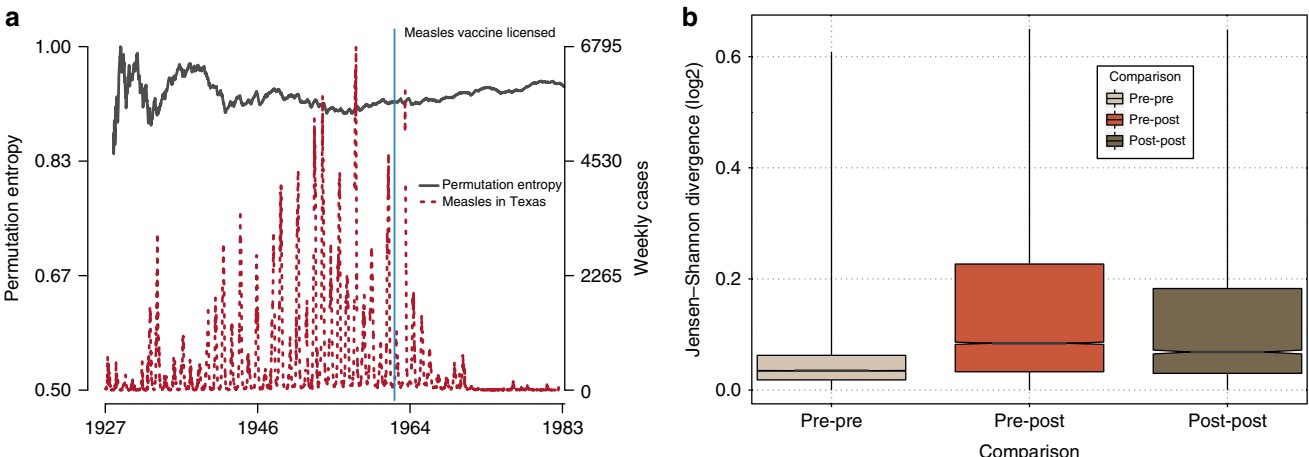

**Fig. 4** Predictability shifts after the start of widespread measles vaccination. **a** Measles in Texas: permutation entropy shifts after the start of widespread vaccination. A rolling window estimate of the permutation entropy for weekly measles cases reported to the CDC from the state of Texas between 1927 and 1983 (red dashed line). The vertical blue line indicates when the first measles vaccine was licensed for use in the United States. Shortly after the vaccine was introduced the permutation entropy increased significantly, which is expected after a system experiences a change to its model structure, in this case vaccination. **b** Permutation entropy detects changing model structure across all states in the United States. The box-plots aggregate the Jensen–Shannon divergences between pairs of windows that are both in the pre-vaccine era (light brown), have one window in the pre- and one in the post-vaccine era (red) and both in the post-vaccine era (dark brown). The Jensen–Shannon divergence of the symbol frequency distribution was calculated between all pairs of rolling 1-year (52 week) windows for measles time series across the United States. The average divergence between symbol distributions was higher between windows in the pre- and post-vaccine era, as compared to windows that were either both in the pre- or both in the post-vaccine era (determined by permutation test and by an analysis of variance that also included state-level differences in the Jensen–Shannon divergence and a Tukey honest significant differences post hoc test, which contained a correction for multiple comparisons). The solid line represents the median, boxes enclose the 25th to 75th percentiles of the distributions, and whiskers cover the entire distribution

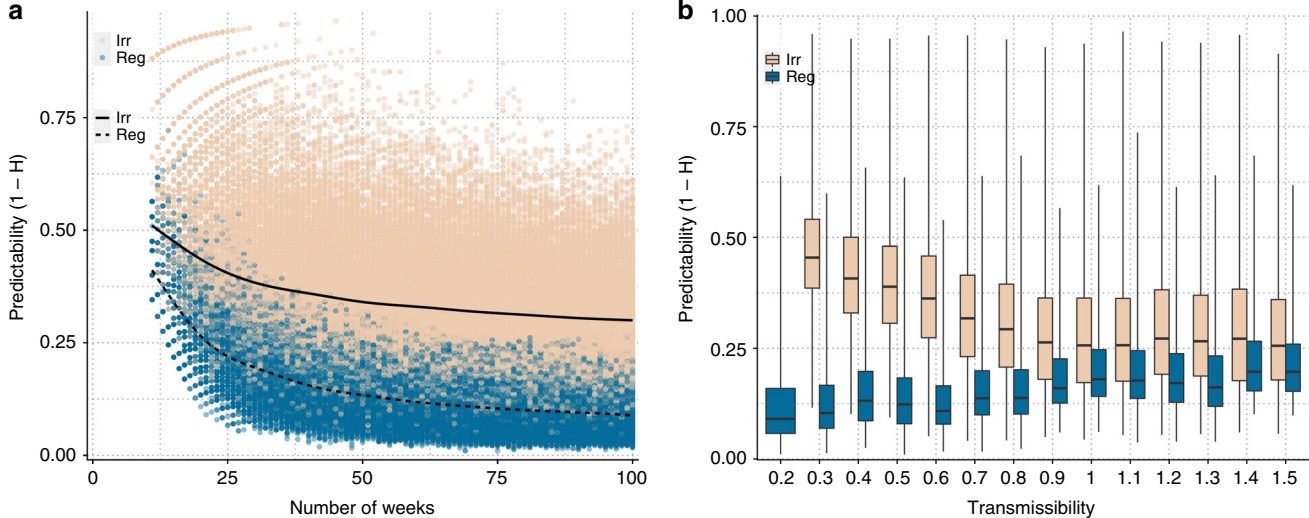

**Fig. 5** Predictability decreases with time-series length and contact homogeneity. **a** For two classes of contact patterns in a synthetic temporal network—regular (reg.), blue and irregular (irr.), brown—we calculate permutation entropy for time series of variable length. For each time-series length we randomly select 1000 starting points and find consistent decreases in the permutation entropy for longer time series. The lines (solid, irr.) and (dashed, reg.) represent the smoothed conditional means, i.e. trend, as determined by fitting a generalized additive model with a thin-plate spline penalty. **b** When considering the uniform (reg., blue) and heterogeneous (irr., brown) scenarios separately, we find that the largest difference in predictability is found below the corresponding epidemic transition consistent with the idea that noisy, sputtering infection trees are harder to predict than the epidemic waves above the transition. The solid line represents the median, boxes enclose the 25th to 75th percentiles of the distributions, and whiskers cover the entire distribution. See the "Epidemic simulations" section in the Supplementary Materials for details on the simulation models

agree with observations that accurate long-range forecasts for complex adaptive systems, e.g., contagions beyond a single outbreak, may be impossible to achieve due to the emergence of entropy barriers. However, they also support the utility and accuracy of dynamical modeling approaches for infectious disease forecasting, especially those that leverage myriad data streams

and are iteratively calibrated as outbreaks evolves. Lastly, our results also suggest that cross-validation over long infectious disease time series cannot guarantee that the correct model for any individual window of time will be favored, which would imply a no free lunch theorem[57] for infectious disease model selection, and perhaps for sociobiological systems more generally.

## Methods

**Permutation entropy**. Here we make use of permutation entropy as a model-independent measure of the growth in complexity and unpredictability of infectious disease time series. Given a time series $\{x_t\}_{t=1,...,N}$ indexed by positive integers, an embedding dimension $d$ and a temporal delay $\tau$, one can consider the set of all sequences of values $s$ of the type $s = \{x_t, x_{t+\tau}, ..., x_{t+(d-1)\tau}\}$. Note that successive values $x_{t+i\tau}, x_{t+(i+1)\tau}$ for generic $i$ can be in an arbitrary relative order. To each $s$, one can associate the permutation $\pi$ of order $d$ that makes $s$ totally ordered, that is $\tilde{d} = \pi(d) = \{x_{t_i}, ..., x_{t_N}\}$ such that $x_{t_i} < x_{t_j} \, \forall t_i < t_j$. In this way, via $\pi$ we associate a rank-order quantity that is independent of the actual values the time series takes and we can associate a probability $p_\pi$ to each permutation by simply counting how many times it appears in the data as compared to the total number of sequences appearing. The permutation entropy of time series $\{x_t\}$ is then given by the Shannon entropy on the permutation orders, that is $H^p_{d,\tau}(\{x_t\}) = -\sum_\pi p_\pi \log p_\pi$. We find that diseases cluster based on the best-fit dimension, $d$ (see Supplementary Figure 2), and that the disease-specific slopes for a random effects model of (log)entropy and (log)time series can be predicted based on the embedding dimension (Supplementary Figure 3).

In the manuscript, we show results obtained by fixing $\tau = 1$ to aid the intuition of the reader and select the most conservative (smallest) value of $H^p(\{x_t\}) = \min_d H^p_{d,\tau=1}(\{x_t\})$ by sweeping over a wide range of possible $d$ values. However, the qualitative results do not change even when we allow for a full sweep across $(d,\tau)$ pairs and setting $H^p(\{x_t\}) = \min_{d,\tau} H^p_{d,\tau}(\{x_t\})$ (see Supplementary Figure 4). In addition, we also confirmed that similar results were obtained by using the weighted permutation entropy, as presented in refs. [32,40] and implemented in the R package statcomp v. 0.0.1.1000[58], see Supplementary Figure 5. Although, it is worth pointing out that weighted permutation entropy attempts to normalize away exactly the kind of structure infectious disease modelers aim to predict.

**Markov chain simulations**. In order to assess the amount of non-random structure in the real outbreak time series, we build synthetic symbolic time series by simulating Markov chains over the symbol distributions obtained from the empirical time series. For each real time series $\{x_t\}_i$, we extract the set of permutation symbols $\{\pi\}$ as in the standard calculation for permutation entropy. We utilize $\tau = 1$ and the embedding dimension $d_i$ previously selected during the permutation entropy computation as described by Brandmeier[38]. For a time series with embedding dimension $d$, there is a maximum number of $d!$ states, corresponding to the possible permutations of length $d$. Using the permutations as states, we then count the number of transitions $n_{ij}$ in the real time series between each pair of symbols $(i,j)$ and use it to build a Markov chain with transition probabilities between states given by $p_{ij} = \frac{n_{ij}}{\sum_j n_{ij}}$. In order to obtain a synthetic symbolic series, we repeatedly start from a randomly selected node and use the Markov Chain described above to produce symbolic series with the same number of symbols as the corresponding real time series. For each iteration, we calculate the associated symbolic entropy. In Supplementary Figure 6, we compare the synthetic entropies versus the permutation entropy of the original time series and show that the former are systematically higher than the real ones, implying that there is additional structure in the outbreak time series that is not captured simply by the probabilistic transition structure.

**Epidemic simulations**. We simulated a standard SIR model with restart on a class of temporal networks in which it is possible to control the expected number of secondary neighbors of nodes. The temporal networks were constructed using the SAD model, a modified version of the well-known activity driven (AD) model[59], in which activations of nodes can involve two (like in the standard AD model) or more nodes establishing reciprocal links. We simulated two types of networks: in the first the number of nodes contacted in every activation was kept constant (regular SAD with $s = 4$); in the second we allowed the number of contacted nodes to fluctuate between interactions (irregular SAD, we sampled $s$ from a normal distribution with mean $\langle s \rangle = 4$ and coefficient of variation $= 0.4$). All networks had $N = 1000$ nodes. Node activities were sampled from a power-law distribution $\sim a^{-\alpha}$ with $\alpha = 2.2$ and rescaled in order to have an average activity $\sim 10^{-2}$, such that nodes activated on average every 100 time steps.

Crucially, for this class of networks it is possible to calculate explicitly the (SIS) critical threshold $\lambda_c = \beta_0/\gamma_0$, where $\beta_0$ and $\gamma_0$ are respectively the matched infection and recovery probabilities at the transition. In order to investigate the behavior of the predictability across the epidemic transition, we fixed $\gamma_0 = 0.1$ and let $\beta$ vary from $0.5\beta_0$ (below the transition) to $4\beta_0$ (far above the transition), where $\beta_0 = \lambda_c\gamma_0$ is the threshold infectivity matching $\gamma_0$. The values of $\gamma_0$ was chosen in order to match the average outbreak peak length to those observed in the data (roughly around 4 weeks). We then simulated the SIR model on the networks described above for $T = 5000$ steps: each outbreak was seeded with five randomly infected nodes and let run its course; at the end of the outbreak, we repeated the seeding until we reached the prescribed time-series length. We calculated the permutation entropy of the synthetic time series in the same way we processed the empirical ones.

**Significance tests on moving-window permutation entropy**. We use a permutation test to determine whether different time-series windows have distinct symbol distributions. Specifically, we fit a multinomial distribution to the normalized symbol frequency distributions and repeatedly simulate data from the estimated multinomials. Then, we calculate the Jensen–Shannon divergence between each pair of simulated distributions. With these simulated distributions, we can ask how often we see fluctuations in our estimate of the permutation entropy just due to sampling. More formally, we use these simulated distributions as a null distribution for calculating a frequentist $p$-value based on the observed Jensen–Shannon divergence between the symbolic frequencies in time series windows.

**Reporting summary**. Further information on experimental design is available in the Nature Research Reporting Summary linked to this article.

**Code availability**. All code associated with this study can be found here: https://github.com/Emergent-Epidemics/infectious_disease_predictability.

## Data availability

Empirical data for all diseases—aside from dengue—were obtained from the U.S.A. National Notifiable Diseases Surveillance System as digitized by Project Tycho[60]. Dengue data were obtained from the Pandemic Prediction and Forecasting Science and Technology Interagency Working Group under the National Science and Technology Council[61]. All data associated with this study can be found here: https://github.com/Emergent-Epidemics/infectious_disease_predictability.

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

## Acknowledgements

We thank Joshua Garland, Pejman Rohani, and Alessandro Vespignani for productive conversations on permutation entropy and helpful comments on an earlier version of the manuscript. S.V.S. received funding support from the University of Vermont and Northeastern University. G.P. received funding support from Fondazione Compagnia San Paolo. S.V.S. and G.P. conducted the study as fellows at IMeRA and drafted the manuscript at Four Corners of the Earth in Burlington Vermont.

## Author contributions

Both authors conceived the project, performed the simulations and calculations, analyzed the empirical data, interpreted the results, and produced the final manuscript.

## Additional information

**Competing interests:** The authors declare no competing interests.

