## [Peer Review File · Nature Communications]

Reviewers' Comments:

Reviewer #1:

Remarks to the Author:

Problem Statement and Results: Forecasting the dynamics of infectious disease outbreaks is a challenging and important problem. There has been a flurry of activity in the recent year on the subject of forecasting infectious disease outbreaks. Researchers have studied a diverse set of methods that range from classical methods in forecasting time series, to the use of sophisticated mechanistic models. Progress has also been made on the data front wherein diverse data sets have been collected and investigated in terms of their utility to forecast infectious disease dynamics. Recent progress has also been made using ensemble techniques to improve the quality of forecasting. Results have primarily been computational in nature and have shown that forecasting infectious disease might often be possible. In contrast to this, very little work has been undertaken to establish the inherent limits on forecasting.

The paper by Scarpino and Petri is a step in this direction. The authors study the information theoretic limits to forecasting infectious disease outbreaks. They use permutation entropy as the method of their choice to study this question. Using diverse time series data on a number of diseases, they investigate this important question.

The paper finds that although forecasting might indeed be feasible, the time horizon over which these forecasts are made depends on both shifting model structure and social network evolution. They conclude that one needs to move beyond simple time series based methods for forecasting and develop dynamic mechanistic models.

Discussion: The paper uses permutation entropy to investigate the fundamental limits of forecasting epidemic dynamics. It can make a potentially important point but needs to be revised substantially before it can be accepted for publication. I have provided suggestions for the same.

1. **Clarifying the definitions:** The level of details provided as regards to how entropy was computed, the Markov chain and the SIR model require substantial details. While some details are provided and I can guess how this is done, it makes it harder to have confidence in my understanding without these details. They are needed since the results critically depend on it. Some examples: what is T , how do you set up τ (was set to 1 as said in the appendix), details on the Markov chain (beyond what is provided). Formal definitions can be given in some cases, in other cases well crafted examples can suffice.

A particular definition that needs clarification is the way you make a time series out of the basic definition of permutation entropy (PE). This is not described. As stated PE is simply a number. Based on the article, this is presumably done by having a sliding window that plots the PE as a time series. More importantly, how does one compute the probability; only within the window or over the entire time series.

2. **Clarifying how models are developed:** The way SIR model was constructed was not clear. At one level this is easy. But there is very little discussion of the probability of transmission which clearly has a role in the overall system dynamics. If one used a random graph model then the rationale for using it is unclear.
3. **Charts, etc:** The captions for the charts etc. should be improved from the standpoint of explaining the results.
4. **4. Substantive comments:**
 - First: comparing the results with results produced by an SIR based network model is good but as they stand need more work. Specifically, one needs to understand the joint interaction of network and transmission probability p .
 - Second the rationale for comparison with a simple random graph style model seems to be unclear. For both the models, a very specific form of network modification is studied to provide a qualitative result that says that model changes might be at play in terms of the observed time series. I think this is a potentially important observation but needs a little more empirical analysis. Why? Because, what ones means by change in model structure is not clear to me.
 - Third, as a way to understand this better, one would like to do the same study using Stochastic difference equations. It will allow one to better understand the effect of network structure.
 - Finally, crucially, disease dynamics with a SIR model on a network do not have the sinusoidal behavior due to the fact that we have nodes recovering after some time and then do not get infected. So understanding the use of the model over long time periods is unclear. Maybe an SIS model would be better suited ? The results with other diseases spanned multiple season; the stylized analysis would be essentially for one season.

Overall Recommendation: The paper contains potentially interesting results. I would like to see a revised version of the manuscript.

Reviewer #2:

Remarks to the Author:

Review of *On the Predictability of Infectious Disease Outbreaks*

=====

==

:author: Samuel V. Scarpino, Giovanni Petri

:date: 14/7/17

In this work the authors construct a framework for quantifying the predictability of infectious diseases in human populations. This makes use of the permutation entropy, a common and powerful tool for quantifying uncertainty in time series without knowledge of the generating mechanism. On the whole, I find that the manuscript needs some clarification before I can recommend it for publication.

Primary Issues

First, it is somewhat disingenuous to say that they are computing the permutation entropy. As they state, they are computing the minimum permutation entropy over a series of stroboscopic samplings of the data. Therefore their predictability measure is of the most predictable sub-sampling of the system, and not of the system as a whole. For example, if a signal consisted of white noise except every 10th value were exactly 0, the predictability would be 1. Therefore it is vastly more accurate to characterize the results of this work as an upper bound on predictability.

Second, in figure 4B the Kullback-Leibler divergence of... something is plotted. To the best of my knowledge this plot is primarily vacuous. It is not stated precisely what pairs of windows are compared and in which order. Due to the asymmetry of the DKL, there is no basis for using it to compare distributions of disease cases at different times. If the goal of this plot is to describe how different the distributions can be at different points within the time series, a better method to quantifying this would be to use the Jensen-Shannon divergence, or perhaps the Earth mover distance.

Lastly, features of figure 4A are tentatively explained through the behavior of an SIR model as seen in figure 5. However, figure 2 demonstrates that the dynamics of the diseases considered here are manifestly dissimilar to that of an SIR model. I find this picking and choosing of aspects of the model to be inappropriate for a manuscript submitted to Nature Communications.

Minor Issues

- When describing the first result, it is stated that windows from "1000 random starting points" are used. It would be more clear to describe it 1000 random starting times, or 1000 random windows into the data.
- R_0 is not defined or described within the text, only described in the caption for fig. 3. A description should appear within the main text.
- In Ref. 14, the weighted permutation entropy is used to discount the contribution of noise to the permutation entropy so that it focuses on the dominant longer scale behavior. I would be interested to see how the results would be modified if the minimum permutation entropy over (τ, d) were changed to the weighted permutation entropy.

Reviewer #3:

Remarks to the Author:

The authors use the concept of permutation entropy to quantify the structural predictability of various infectious disease dynamics based on historical data. The main findings of the study are that the predictability depends upon the time horizon, the type of disease, and that the model structure and the structure of social networking explain most of the differences in predictability. Furthermore the

authors stretch the need for developing efficient dynamic modeling approaches that will enhance our forecasting capability.

The manuscript reads well and is generally well structured. It targets on a yet unsolved problem that of the efficient prediction of infectious disease dynamics.

Having said that, I do not see the novelty in the methodology or in the main findings of the study that would support a publication in Nature Communications.

On the one hand, the use of permutation entropy for the quantification of predictability of time series is not a new idea. It was introduced by Banadt and Pompe in 2002 and since then there are numerous applications of this concept. On the other hand, the main findings are already known in the community: the forecasting capability is limited by the time horizon, the predictability depends on the heterogeneity of the network (see e.g. Hafnagel et al., 2003, PNAS, 101, 15124–15129).

Some other minor and major points:

1. Figure 2: the color coding is rather vague. The authors should consider using symbols like stars triangles etc. The same holds also for the confidence intervals of predictability (at what level?) which are not distinguishable.
2. Figure 2: For the case of Zika (?), in contrast to the other infectious diseases, it seems that there is an "optimal" value of the predicting horizon that is followed with its sharp decrease. The authors do not (should) comment on this in the text.
3. For many systems the permutation entropy has been shown to be equivalent to the Kolmogorov-Sinai entropy. A comparison between the two metrics would enhance the presentation of the work.
4. It would be interesting to show a comparison between the permutation entropy and the weighted permutation entropy (Fadlallah, et al. ,2013, Phys. Rev. E 87, 02291).
5. The analysis is post-hoc. For the implementation of the proposed methodology in real emerging situations the values of the two parameters, i.e. the embedding dimension and the embedding delay are not known a-priori and the analysis of past data does not guarantee their validity for the future. Mutations of the virus, changes in contact transmission network etc may change the embedding parameters and hence our forecasting ability.
6. The authors highlight the need for constructing efficient dynamic network models. However they don't present how in a systematic way how one can construct this link between the concept of permutation entropy and dynamic models.
7. The authors have to present better the SIR model and its parameters for the results to be reproducible. Also the algorithm of the network construction is missing.

1
2
3
4
5
6

Response to reviewer #1

1.) The level of details provided as regards to how entropy was computed, the Markov chain and the SIR model require substantial details. While some details are provided and I can guess how this is done, it makes it harder to have confidence in my understanding without these details. They are needed since the results critically depend on it. Some examples: what is T , how do you set up τ (was set to 1 as said in the appendix), details on the Markov chain (beyond what is provided). Formal definitions can be given in some cases, in other cases well crafted examples can suffice. A particular definition that needs clarification is the way you make a time series out of the basic definition of permutation entropy (PE). This is not described. As stated PE is simply a number. Based on the article, this is presumably done by having a sliding window that plots the PE as a time series. More importantly, how does one compute the probability; only within the window or over the entire time series.

We have edited our description of the methods to improve interpretability and more clearly define the key parameters. Regarding the amount of data used in calculating the permutation entropy, we took three different approaches, all of which provide qualitatively similar conclusions. First, in figures 1 and 4, we used non-overlapping windows of size $N = 52$ weeks; second, in figure 2, we selected 1,000 random starting locations for each state-level time series for each disease and calculated the permutation entropy in rolling windows between 2 - 104 weeks; and finally, in figure 3 we used the entirety of each of the state-level time series. We chose to present the results using three different kinds of “windows” to demonstrate that our findings weren’t dependent on how we estimated the permutation entropy.

We also provide the additional details on the Markov chain in the supplement, in particular we describe how we constructed the Markov chain transition probabilities and the synthetic timeseries from the original ones.

Lastly, we’ve modified the following paragraph in the methods to more fully define the key parameters associated with each permutation entropy calculation.

“As described above, calculating the permutation entropy of a time series requires selecting values for the embedding dimension d , the time delay τ , and the window length N . In this study, our goal was to find conservative values of H^p by searching over a wide

range of possible (d, τ) pairs and setting $H^p(\{x_t\}) = \min_{d, \tau} H_{d, \tau}^p(\{x_t\})$. However, the value of H^p should always decline as the embedding dimension d grows, i.e. no minimum value of H^p will exist for finite windows sizes N . To address this issue, we follow Brandmaier (2015) and exclude all unobserved symbols when calculating H^p , which acts as a “penalty” against higher dimensions and results in a minimum value of H^p for finite length time series. To control for differences in dimension and for the effect of time series length on the entropy estimation, we normalize the entropy by $\log(d!)$, ensure that each window is greater in length than $d!$, and confirm that the estimate of H^p has stabilized (specifically that the marginal change in H^p as data are added is less than 1%). To facilitate interpretation, we present results from continuous intervals by fixing $\tau = 1$. However, our results generalize to the case where we fix both d and τ across all diseases and where we minimize over a range of (d, τ) pairs (see Supplement).”

2.) The way SIR model was constructed was not clear. At one level this is easy. But there is very little discussion of the probability of transmission which clearly has a role in the overall system dynamics. If one used a random graph model then the rationale for using it is unclear.

We agree with the Reviewer that the previous description of the SIR process we simulated was unclear. We added a section to the supplementary materials detailing the process and implementation of these simulations. Briefly, we considered temporal networks built using a recent network model where it is possible to specify the extent of the interaction among second neighbours and calculate analytically the epidemic threshold. We simulated the spreading process for two cases: one in which the number of interactions between second neighbors is fixed, and a second in which the number of second neighbour interactions fluctuates in time. For both cases, we simulated the spreading process for a range of parameters from far below to far above the epidemic threshold ($.5\lambda_c$ to $4\lambda_c$).

3.) The captions for the charts etc. should be improved from the standpoint of explaining the results..

We have carefully edited to captions to improve their utility when interpreting our results.

4.) Comparing the results with results produced by an SIR based network model is good but as they stand need more work. Specifically, one needs to understand the joint interaction of network and transmission probability p .

We refer to the reply to point 2 above. We would like to further stress that for the temporal network simulations, we found higher predictability for all infection probabilities in the case of networks displaying heterogeneity in the number of contacts. However, the largest difference between the regular and irregular contact temporal networks is found below the transition. This is consistent with the idea that, above the transition, outbreaks behave essentially as quasi-deterministic waves and the effect of the underlying structure is limited; below the transition instead, outbreaks display much more fragmented and complex evolution resulting in lower predictability for networks with uniform structure and higher predictability for networks with strong heterogeneities (similar to modeling results obtained by Hafnagel et al. (2003), as discussed below and in the revised version of our manuscript).

5.) Second the rationale for comparison with a simple random graph style model seems to be unclear. For both the models, a very specific form of network modification is studied to provide a qualitative result that says that model changes might be at play in terms of the observed time series. I think this is a potentially important observation but needs a little more empirical analysis. Why? Because, what ones means by change in model structure is not clear to me.

We did not intend our simulations to be an exhaustive summary of the potential origins of different patterns of predictability, even in very similar systems. Rather, we want to show that even toy synthetic systems perturbed at the structural (more or less heterogenous contact patterns) or dynamical level (parameter drift) can already show alterations in their predictability. In addition, our goal for the SIR curve in Figure 2 was to provide a reference point for readers evaluating the empirical data, which is the primary result presented in that figure.

6.) Third, as a way to understand this better, one would like to do the same study using Stochastic difference equations. It will allow one to better understand the effect of network structure.

We agree that an analytical approach to studying the permutation entropy of disease transmission on networks might provide novel insight into the information theoretic dynamics of outbreak predictability. However, given the complexity of such an endeavor, we feel it is beyond the scope of our current analysis. And, after conversations with experts in this area, we believe it would require substantial analytical advancements beyond what has currently been published. We would be very happy to collaborate with the reviewer on a future study if this is their area of expertise.

7.) Finally, crucially, disease dynamics with a SIR model on a network do not have the sinusoidal behavior due to the fact that we have nodes recovering after some time and then do not get infected. So understanding the use of the model over long time periods is unclear. Maybe an SIS model would be better suited? The results with other diseases spanned multiple seasons; the stylized analysis would be essentially for one season.

We agree with the reviewer that our previous text was unclear. We implemented an SIR model with restart after each outbreak, which allows to simulate multiple seasons and repeated outbreaks. We provide an improved description of the simulations in the Supplement.

Response to reviewer #2

1.) First, it is somewhat disingenuous to say that they are computing the permutation entropy. As they state, they are computing the minimum permutation entropy over a series of stroboscopic samplings of the data. Therefore their predictability measure is of the most predictable sub-sampling of the system, and not of the system as a whole. For example, if a signal consisted of white noise except every 10th value were exactly 0, the predictability would be 1. Therefore it is vastly more accurate to characterize the results of this work as an upper bound on predictability.

The Reviewer is correct that there are many other dimensions one could use as the embedding dimensions, which would likely result in different predictabilities. Trivially, without penalizing the entropy calculation, for a finite-length time series the entropy will always decrease with increasing dimension. The issue of decreasing entropy with increasing dimension is why we consider the choice of the minimum entropy dimension as a strength of our methodology. Specifically, our approach allows one to identify the size of the blocks over which the maximum predictability can be expected. Additionally, and as detailed in our response to Reviewer 1, we performed three distinct kinds of data sampling, using two variations on the permutation entropy (weighted and unweighted) and obtained the same qualitative results.

Regarding the reviewer's comment on a white noise time series where every 10th value was zero, this example is actually a bit tricky (making it a very interesting case to consider). If the mean of the white noise was zero and the variance was small, e.g., $\mu = 0, sd = 1$, then the predictability for a white noise time series with every 10th value set to 0 would be equivalent to a white noise time series. This is because the time series would be equally likely to be above or below the 10th value, meaning the entropy over the symbols would be equivalent. If instead, we set the 10th value to 100 (or -100), but kept the white noise at $\mu = 0, sd = 1$, then the predictability would be higher (because the time series would almost certainly be above -100 or below 100). We demonstrate this in the figure below.

2.) Second, in figure 4B the Kullback-Leibler divergence of... something is plotted. To the best of my knowledge this plot is primarily vacuous. It is not stated precisely what pairs of windows are compared and in which order. Due to the asymmetry of the DKL, there is no basis for using it to compare distributions of disease cases at different times. If the goal of this plot is to describe how different the distributions can be at different points within the time series, a better method to quantifying this would be to use the Jensen-Shannon divergence, or perhaps the Earth mover distance.

We agree with the reviewer that using the KL divergence was unclear and led to ambiguities in the comparison. We replaced it with the Jensen-Shannon divergence as suggested by the reviewer and found qualitatively similar results.

3.) Lastly, features of figure 4A are tentatively explained through the behavior of an SIR model as seen in figure 5. However, figure 2 demonstrates that the dynamics of the diseases considered here are manifestly dissimilar to that of an SIR model. I find this picking and choosing of aspects of the model to be inappropriate for a manuscript submitted to Nature Communications.

We refer to the reply to point 2 of Reviewer 1. We intended the simulations in figure 2 to be a guide for the explanation and a toy environment to show potential mechanistic sources of reduced predictability. For Figure 5, we have substantially expanded our investigation of the simulation results and again direct the reviewer to our earlier responses.

4.) When describing the first result, it is stated that windows from "1000 random starting points" are used. It would be more clear to describe it 1000 random starting times, or 1000 random windows into the data.

We have modified the text to improve the interpretability of this result. This sentence now reads, "Focusing first on the predictability over short timescales (Figure 2), for each time series we average H^p over temporal windows of width up to 100 weeks by selecting 1000 random starting points from each state-level time series for disease and calculating H^p for windows of length 10, 12, ..., 100."

5.) R_0 is not defined or described within the text, only described in the caption for fig. 3. A description should appear within the main text.

We have added the following definition after our first use of R_0 in the main text, “After re-normalizing time for each disease by its corresponding R_0 (the average number of secondary infections a pathogen will generate during an outbreak/epidemic when the entire population is susceptible, very large, and is seeded with a single infectious individual).”

6.) In Ref. 14, the weighted permutation entropy is used to discount the contribution of noise to the permutation entropy so that it focuses on the dominant longer scale behavior. I would be interested to see how the results would be modified if the minimum permutation entropy over (τ, d) were changed to the weighted permutation entropy.

We thank the reviewer for the suggestion. We performed the weighted entropy analysis and found that the results do not change qualitatively as compared to the case of standard permutation entropy, which in fact makes our results even stronger as it confirms that the predictability decreases we observe genuinely stem from the alteration of the ordinal patterns rather than from scale effects related to the size of the outbreak. These results are contained in the Supplement because our position is that most researchers are focused on predicting the unweighted epidemic curves; however, if the reviewer disagrees, we would be happy to move the figure describing these results to the main text.

Response to reviewer #3

1.) The manuscript reads well and is generally well structured. It targets on a yet unsolved problem that of the efficient prediction of infectious disease dynamics. Having said that, I do not see the novelty in the methodology or in the main findings of the study that would support a publication in Nature Communications.

As the reviewer points out in subsequent comments, we are not the first to argue for the presence of entropy barriers to infectious disease prediction nor are we the first to apply permutation entropy to the question of time series predictability. However, we are the first to demonstrate the presence of entropy barriers to disease forecasting in a broad, comparative framework and to begin evaluating plausible mechanisms for the emergence of such barriers against empirical data. In particular, our result that permutation entropy is able to distinguish pre- and post-vaccine eras for measles time series is, to the best of our knowledge, the first demonstration of shifting model structure being detected with permutation entropy for any ecological time series. It’s also worth noting that the proposed methodology for selecting optimal embedding dimensions, which was a critical barrier to using permutation entropy,

FIG. 1. Here, we simulated four different time series and calculate the predictability (1-H) from 1,000 random starting points in each time series. The first, white noise (light gray) contains 5000 random numbers drawn from a Gaussian distribution with $\mu = 0$, $sd = 1$. The remaining three time series contain exactly the same random numbers as in the white noise time series, i.e. we do not resample, but every 10th value is set to either -100 (blue), 0 (black), or 100 (red).

and the results demonstrating that permutation entropy was associated with established metrics for predictability and entropy in dynamics systems were both published in 2015. Therefore, we argue that only recently have the tools and theoretical foundations been in place for a study such as ours.

In addition, we show that all the diseases we study, once time-renormalized on their reproductive number, display a universal behaviour in the scaling of predictability with availability of data. Another crucial observation is that while this universal behaviour is common across diseases, what is not common is the heterogeneity across different temporal windows for different time series signaling, which again we believe is a genuine novel contribution.

2.) On the one hand, the use of permutation entropy for the quantification of predictability of time series is not a new idea. It was introduced by Banadt and Pompe in 2002 and since then there are numerous applications of this concept. On the other hand, the main findings are already known in the community: the forecasting capability is limited by the time horizon, the predictability depends on the heterogeneity of the network (see e.g. Hafnagel et al., 2003, PNAS, 101, 15124-15129).

We agree with the Reviewer that heterogeneity has a known effect on the predictability horizon of disease spreading. However, to-date the evidence for the effect of heterogeneity has either come from simulation studies or from studies dependent on a particular model structure. Our work advances from these early efforts by identifying barriers using a model-independent approach, i.e. permutation entropy, and by doing so in a comparative framework. Specifically, we observe that the permutation entropy rate production can be described in terms of two distinct parameters, one disease-specific (R_0), the other emerging dynamically from our optimization of the permutation entropy embedding dimension, which then provides a data-driven predictability horizon. Further, we show that it is not only structural heterogeneity but also dynamical and biological variability that can result in decreased predictability (e.g. the vaccination example and the temporal network simulations).

We also agree that citing Hafnagel et al., 2003, along with a number of other more recent demonstrations of entropy barriers, is appropriate and we have added the following paragraph to the discussion:

Our finding, that horizons exist for infectious disease forecast accuracy and that aggregating over multiple outbreaks can actually decrease predictability is supported by five additional lines of evidence. First, Hufnagel et al. (2004), using data on the 2004 SARS outbreak and airline travel networks, demonstrated that heterogeneity in connectivity can improve predictability [1]. Second, Domenech de Cellés et al. (2018) noted a sharp horizon in forecast accuracy for whooping cough outbreaks in Massachusetts, USA [2]. Third, Coletti et al. (2018) demonstrated that seasonal outbreaks of influenza in France often have unique spatiotemporal patterns, some of which cannot be explained by viral strain, climate, nor commuting patterns [3]. Fourth, Artois et al. (2018) found that while it was possible to predict the presence human A(H7N9) cases in China, they were unable to derive accurate forecasts for the temporal dynamics of human case counts [4]. Finally, using state-level data from Mexico on measles, mumps, rubella, varicella, scarlet fever and pertussis, Mahmud et al. (2017) showed evidence that while short-term forecasts were often highly accurate, long-term forecast quality quickly degraded [5].

3.) Figure 2: the color coding is rather vague. The authors should consider using symbols like stars triangles etc. The same holds also for the confidence intervals of predictability (at what level?) which are not distinguishable.

We added a sentence clarifying whether the confidence intervals overlap. Although we investigated the addition of symbols, the graph became overly cluttered.

4.) Figure 2: For the case of Zika (?), in contrast to the other infectious diseases, it seems that there is an optimal value of the predicting horizon that is followed with its sharp decrease. The authors do not (should) comment on this in the text.

We have replaced Zika with data on dengue in Puerto Rico. The Zika data were from a single outbreak at the country-level, which makes it difficult to interpret differences between Zika and the other diseases. We agree that the potential for differences in predictability across levels of geographic aggregation and between diseases that are vectored vs. transmitted from humans-to-humans are important areas of future research. Nevertheless, we feel they are beyond-the-scope of our current investigation.

5.) For many systems the permutation entropy has been shown to be equivalent to the Kolmogorov-Sinai entropy. A comparison between the two metrics would enhance the presentation of the work.

We thank the Reviewer for this comment. We agree that permutation entropy (or a slightly generalized version [6–8]) has been shown to be equivalent to the KS entropy in some systems, and to provide an upper bound on the permutation entropy in others. However, in the case of discrete timeseries adopting permutation entropy is significantly easier and more parsimonious to compute.

6.) *It would be interesting to show a comparison between the permutation entropy and the weighted permutation entropy (Fadlallah, et al. ,2013, Phys. Rev. E 87, 02291).*

We repeated the analysis with weighted entropy and found that the results are confirmed also in the weighted case. Please refer to the reply to point 6 of Reviewer 2.

7.) *The analysis is post-hoc. For the implementation of the proposed methodology in real emerging situations the values of the two parameters, i.e. the embedding dimension and the embedding delay are not known a-priori and the analysis of past data does not guarantee their validity for the future. Mutations of the virus, changes in contact transmission network etc may change the embedding parameters and hence our forecasting ability.*

We agree with this point, and, in fact, this was a key motivation for our study. To date, many researchers focused on infectious disease prediction have not appreciated that one must both identify an optimal window for forecasting and that this window is likely to change from outbreak-to-outbreak and from disease-to-disease. We believe that perhaps the key practical outcome from our investigation is to make people aware that the embedding dimension of the forecast is a key parameter for evaluating the quality of a disease forecast and that estimating it from historical data cannot guarantee that the same embedding dimension will generalize to future outbreaks.

8.) *The authors highlight the need for constructing efficient dynamic network models. However they dont present how in a systematic way how one can construct this link between the concept of permutation entropy and dynamic models.*

The reviewer is correct that the simulations we presented are not a systematic analysis of the interaction between dynamic models, dynamic network structure and their effect on the predictability of the resulting outbreaks. The rationale for our simulations was to show one possible network mechanism that is able to produce the observed change in predictability even in very simplified systems. The same argument also supports our example of model shift due to vaccination. We agree with the reviewer that a complete characterization is interesting (and a few steps have been recently taken, for example in [9]), but we consider it beyond the scope of this paper, where the focus is instead on the empirical limits to outbreak prediction.

9.) *The authors have to present better the SIR model and its parameters for the results to be reproducible. Also the algorithm of the network construction is missing.*

We added a section in the SI with an improved the description of how the networks were constructed and the simulations performed.

-
- [1] Lars Hufnagel, Dirk Brockmann, and Theo Geisel. Forecast and control of epidemics in a globalized world. *Proceedings of the National Academy of Sciences of the United States of America*, 101(42):15124–15129, 2004.
 - [2] Matthieu Domenech de Cellès, Felicia MG Magpantay, Aaron A King, and Pejman Rohani. The impact of past vaccination coverage and immunity on pertussis resurgence. *Science translational medicine*, 10(434):eaaj1748, 2018.
 - [3] Pietro Coletti, Chiara Poletto, Clement Turbelin, Thierry Blanchon, and Vittoria Colizza. Shifting patterns of seasonal influenza epidemics. *Scientific Reports*, 8(12786):1–12, 2018.
 - [4] Jean Artois, Hui Jiang, Xiling Wang, Ying Qin, Morgan Percy, Shengjie Lai, Yujing Shi, Juanjuan Zhang, Zhibin Peng, Jiandong Zheng, et al. Changing geographic patterns and risk factors for avian influenza a (h7n9) infections in humans, china. *Emerging infectious diseases*, 24(1):87, 2018.
 - [5] AS Mahmud, CJE Metcalf, and BT Grenfell. Comparative dynamics, seasonality in transmission, and predictability of childhood infections in mexico. *Epidemiology & Infection*, 145(3):607–625, 2017.
 - [6] José M Amigó. The equality of kolmogorov–sinai entropy and metric permutation entropy generalized. *Physica D: Nonlinear Phenomena*, 241(7):789–793, 2012.
 - [7] Karsten Keller and Mathieu Sinn. Kolmogorov–sinai entropy from the ordinal viewpoint. *Physica D: Nonlinear Phenomena*, 239(12):997–1000, 2010.
 - [8] Christoph Bandt, Gerhard Keller, and Bernd Pompe. Entropy of interval maps via permutations. *Nonlinearity*, 15(5):1595, 2002.
 - [9] Frank Pennkamp, Alison Iles, Joshua Garland, Georgina Brennan, Ulrich Brose, Ursula Gaedke, Ute Jacob, Pavel Kratina, Blake Matthews, Stephan Munch, et al. The intrinsic predictability of ecological time series and its potential to guide forecasting. *bioRxiv*, page 350017, 2018.

Reviewers' Comments:

Reviewer #1:

Remarks to the Author:

The revised paper is substantially improved. I do have a few technical questions/comments:

- 1) When computing PE, you have τ and d . A simple example should be worked out in the appendix.
- 2) Am still not completely convinced with the SIR dynamics using a temporal network. It still seems like a very specific way of constructing temporal networks (it is clearly better than the first version).
- 3) In several places, I feel that an explicit description of the method or the analytical form needs to be given, the description in plain english is useful but needs to be complemented with a formal description.
- 4) Plots showing $1-H$ are way too busy making them hard to understand. Is there another way to depict this information. Maybe discretize it at some chosen time points in addition (along with uncertainty) would be perhaps easier?
- 5) I like the weighted entropy measure and is clearly more appropriate.

Reviewer #2:

Remarks to the Author:

I am satisfied with the modified manuscript.

Reviewer #3:

Remarks to the Author:

The authors have responded adequately to my remarks. Hence I recommend its acceptance.

Reviewer #1 (Remarks to the Author):

The revised paper is substantially improved.

We thank the reviewer for their opinion and for the help in improving the manuscript.

I do have a few technical questions/comments:

1) When computing PE, you have τ and d . A simple example should be worked out in the appendix.

We added a figure to the supplement with the explicit calculation of the permutation entropy.

2) Am still not completely convinced with the SIR dynamics using a temporal network. It still seems like a very specific way of constructing temporal networks (it is clearly better than the first version).

We agree that we adopted a specific example; however, our intent was to show that it was possible to produce shifts in observed predictability even in simple synthetic examples, rather than provide an extensive description of all the factors that might affect the predictability. As we discuss in the paper, this remains an important area of future research.

3) In several places, I feel that an explicit description of the method or the analytical form needs to be given, the description in plain english is useful but needs to be complemented with a formal description.

We provide analytical descriptions of all terms throughout the manuscript.

4) Plots showing 1-H are way too busy making them hard to understand. Is there another way to depict this information. Maybe discretize it at some chosen time points in addition (along with uncertainty) would be perhaps easier?

We have added a panel to figure 2 discretizing the permutation entropy after four months.

5) I like the weighted entropy measure and is clearly more appropriate.

In our opinion, many researchers engaged in predicting outbreaks and public health officials studying outbreaks focus on the raw time series and, therefore, are interested in the non-weighted version of the permutation entropy. We believe it remains most appropriate to include both measures (as we have done in this version).

Reviewer #2 (Remarks to the Author):

I am satisfied with the modified manuscript.

We thank the reviewer for their opinion and for the help in improving the manuscript.

Reviewer #3 (Remarks to the Author):

The authors have responded adequately to my remarks. Hence I recommend its acceptance.

We thank the reviewer for their opinion and for the help in improving the manuscript.